# Improving Radar Human Activity Classification Using Synthetic Data with Image Transformation

**DOI:** 10.3390/s22041519

**Published:** 2022-02-16

**Authors:** Rodrigo Hernangómez, Tristan Visentin, Lorenzo Servadei, Hamid Khodabakhshandeh, Sławomir Stańczak

**Affiliations:** 1Fraunhofer Heinrich Hertz Institute, 10587 Berlin, Germany; tristan.visentin@hhi.fraunhofer.de (T.V.); hamid.khb.89@gmail.com (H.K.); slawomir.stanczak@hhi.fraunhofer.de (S.S.); 2Infineon Technologies AG, 85579 Munich, Germany; lorenzo.servadei@infineon.com; 3Department of Electrical and Computer Engineering, Technical University of Munich, 80333 Munich, Germany; 4Faculty IV, Electrical Engineering and Computer Science, Technical University of Berlin, 10587 Berlin, Germany

**Keywords:** radar, machine learning, deep learning, human activity classification, image transformation, domain shift

## Abstract

Machine Learning (ML) methods have become state of the art in radar signal processing, particularly for classification tasks (e.g., of different human activities). Radar classification can be tedious to implement, though, due to the limited size and diversity of the source dataset, i.e., the data measured once for initial training of the Machine Learning algorithms. In this work, we introduce the algorithm Radar Activity Classification with Perceptual Image Transformation (RACPIT), which increases the accuracy of human activity classification while lowering the dependency on limited source data. In doing so, we focus on the augmentation of the dataset by synthetic data. We use a human radar reflection model based on the captured motion of the test subjects performing activities in the source dataset, which we recorded with a video camera. As the synthetic data generated by this model still deviates too much from the original radar data, we implement an image transformation network to bring real data close to their synthetic counterpart. We leverage these artificially generated data to train a Convolutional Neural Network for activity classification. We found that by using our approach, the classification accuracy could be increased by up to 20%, without the need of collecting more real data.

## 1. Introduction

### 1.1. Motivation

The fast advances in Machine Learning(ML) research help to solve more and more important problems. Lately, sophisticated ML methods found their way into radar signal processing and analysis [1,2,3]. However, generating high-quality datasets that adequately represent reality to its full extent is still highly demanding. Methods like domain adaptation, transfer learning and Generative Adversarial Networks (GANs) assist the improvement of classification or regression tasks from incomplete datasets [4]. In this paper, we focus on human activity classification with radar due to the research interest that it has currently awakened [5,6,7,8], for which we count on the help of millimeter-wave radar sensors operating at 60 GHz. Problems with datasets in this context can originate from several reasons:Incomplete dataset, meaning that a limited number of subjects executed the activities.Insufficient measurement time, meaning that just a short time frame of the activity for each person is captured for the dataset.Different radar sensor settings or parameters, e.g., bandwidth, measurement time, repetition time, etc.Inconsistent measurement times, meaning the same person performing a specific activity task executes the activity differently at different times, e.g., after a coffee break or at a different distance to the sensor, etc.

From these given problem examples, 1 and 2 can only be solved adequately by gathering more data of different people for a longer time frame. Problems 3 and 4 are promising cases to be investigated though, as they lend themselves to be solved by ML methods. Our following evaluation showed that the same data being measured with equal radar sensors but different settings (example 3) did not lead to a large decline in classification accuracy [9]. Thus, in this paper, we focus on problem 4.

In the dataset originating from our measurement campaign (outlined in [9]), the data also showed a statistical discrepancy for different measurement time intervals (different subjects, disparate activity execution, etc.). Our idea to tackle this issue is to investigate whether we can utilize synthetic data to enhance our dataset. The proposed method uses a radar reflection model of the recorded subject that is based on a human motion model (also referred to as kinematic model) extracted from stereo video camera data. This was possible through the simultaneous recording of the activities with radar sensors and such cameras. Thus, this motion model provides the information to generate radar data from moving humans outlined by Chen [10]. Using this common model, we obtain synthetic radar data to augment our dataset. Furthermore, this approach offers the chance to augment the real radar data with synthetic data originating from human motion data created somewhere else and/or in future measurements. This makes the radar model universally applicable.

In this paper, we introduce an approach that we name Radar Activity Classification with Perceptual Image Transformation (RACPIT), proceeding in the following way:We first show that data from a source time and a target time interval in our dataset differ significantly.As a baseline, we train a Convolutional Neural Network (CNN) with only real data from one source time interval and test the trained CNN with data from a target time interval.We demonstrate the approach of perceptual loss with Image Transformation Networks [11] to show that we can increase classification accuracy by only using synthetic radar data from the target time (generated by taking the human motion data from the target time and using it as input for the human radar reflection model from Chen [10]).We propose improvements of this method for future research.

### 1.2. Related Work

Among ML algorithms, deep learning has gained popularity over the years as a technique to classify human activities using radar features [5,6,9,12,13,14]. High-quality public radar data is still hardly available, however, despite the great efforts that exist in this regard [15]. Yeo et al. [16] and Zhang et al. [17] have gathered remarkable datasets for tangible interaction and road object detection, respectively; in both cases, they have opted for Frequency-Modulated Continuous-Wave (FMCW) radar to retrieve information about the targets’ range, velocity, or angle. In the area of human activity classification, Sevgi Z. Gurbuz et al. [14] have collected a comprehensive dataset that has been acquired with FMCW and Ultra-wideband (UWB) radar sensors for different frequency bands. Nevertheless, it remains unclear whether their FMCW data allows range detection.

Due to this limitation in data availability, some authors resort to simulated data to train their deep-learning algorithms [7,8]. The core of the problem is then shifted to the choice of a suitable model to simulate human motion; this can be either analytically tailored to specific movements [18] or rely on MOtion CAPture (MOCAP) data [19].

Regarding the different methods to bridge the discrepancies between real and simulated radar data, visual domain adaptation techniques constitute a sensible choice [7,8]. In this work, however, we focus on ML-aided image-to-image translation. Among the different approaches of image-to-image translation in the Computer Vision (CV) community, GANs remains one of the most popular ones [20,21]. Originally conceived for style transfer, perceptual-loss approaches have emerged in recent years as an alternative strategy to tackle image-to-image translation, especially in scenarios such as that of super-resolution images [11]. Here the image translation focuses on the upsampling of a low-quality picture into a bigger, high-definition version of it.

## 2. Radar Sensor and Dataset

We use the dataset from [9], which consists of the following five human activities:(a)Standing in front of the sensor.(b)Waving with the hand at the sensor.(c)Walking back and forth to the sensor.(d)Boxing while standing still.(e)Boxing while walking towards the sensor.

In other words, the considered activities belong to a set A with $\left|A\right|=5$. For each activity, data were acquired simultaneously with four FMCW radar sensors in a series of non-contiguous recordings. The duration of a single recording ranges from 30 s to 2 min and the recorded data per activity adds up to roughly 10 min, with an overall number of recordings per sensor of M=69. In this recording environment that can be seen in Figure 1, two male adults took turns to perform the activities individually.

All sensors are based on Infineon’s BGT60TR13C chipset, a 60 GHz FMCW radar system (c.f. Figure 1b). Each sensor was previously configured individually with one of the four different settings (I–IV) listed in Table 1.

Since each activity was recorded for around 10 min, we collected up to 50 min of data per configuration. These 50 min translate to around 60,000 or 93,000 frames depending whether the configuration’s frame period is 50 ms or 32 ms, respectively. We express the relation between the total number of frames per configuration, *L*, with the number of recordings per configuration, M, through the following expression:(1)$$L=l_{1}+l_{2}+\cdots +l_{M}\;,$$
where $l_{i}$ for $i\in \left\{1,\;\ldots \;,M\right\}$ equals the number of frames for the *i*-th recording.

Preliminary data exploration has shown statistical and qualitative differences between recordings that lead to poor generalization across them. In order to explore this phenomenon, we split our recordings into a source domain S of length $L_{S}$ frames and a target domain T of length $L_{T}$ frames. We do so by assigning the first *m* recordings to S and the last M−m recordings to T. The splitting point *m* is chosen so that:(2)$$L_{S}=l_{1}+\cdots +l_{m}≃L_{T}=l_{m+1}+\cdots +l_{M}≃L/2\;.$$

The FMCW sensors emit a train of $n_{c}$ linear frequency-modulated electromagnetic pulses (so-called chirps), as outlined in Figure 2. The reflected and returned signal is then mixed down with the emitted one to an Intermediate Frequency (IF) and sampled with $n_{s}$ samples per chirp. In that way, we obtain the raw data, which we further rearrange in an $n_{c}\times n_{s}$ matrix called *frame*. On a frame, we refer to the horizontal direction ranging from 1 to $n_{s}$ as the *fast time* in contrast to the vertical direction from 1 to $n_{c}$, which we refer to as *slow time*. Frames are acquired periodically at a fixed frame rate and pre-processed with the following steps, which revolve around the Fast Fourier Transform (FFT) algorithm:Moving Target Indication (MTI): In order to filter out the clutter from static objects, we calculate the mean across the slow time to obtain an $n_{s}$-long vector that we subtract from every column of the frame [22].Range FFT: We perform an FFT across the fast time for every chirp, obtaining thus $n_{c}$
*range profiles* [22]. Prior to that, we apply a Hanning window [23] and zero-padding, so that the range profiles have a length of 128 regardless of the value of $n_{s}$.Doppler FFT: We also perform an FFT across the slow time, turning the range profiles into a Range-Doppler Map (RDM) that conveys information on the returned power across range and Doppler, i.e., distance and velocity (c.f. Figure 3c). Here we use a Chebyshev window with 60 dB sidelobe attenuation [23] and the same zero-padding strategy as for the range FFT. As a result, the dimensions of the RDMs are 128×128 for all configurations.Spectrogram extraction: Similar to [12], we stack the pre-processed frames into an RDM sequence and we sum this over the Doppler and range axes to get range and Doppler spectrograms, respectively (c.f. Figure 4). Here the magnitude of both spectrograms is converted to decibels.Slicing and normalization: As the last step, we slice each long recording into 64-frame long spectrograms with an overlap such that the last 56 frames of one spectrogram coincide with the first 56 frames of the next sliced spectrogram. Furthermore, we shift the decibel values of each sliced spectrogram so that the maximum value equals 0 dB and we subsequently clip out all values below −40 dB.

As a result of this preprocessing and the domain split described in Equation (Equation 2), datasets comprise between 3000 and 6000 data points, depending on the configuration. A single data point corresponds to a tuple containing one range spectrogram and one Doppler spectrogram; the spectrogram size in either case is M×N for M=64 and N=128.

### Data Synthesis

For the measuring campaign, we have also recorded video data from four different cameras. Moreover, we have extracted the pose of the subject from the video data using Detectron2, a CV library provided by Meta Platforms Inc. (formerly Facebook Inc.) [24]. The extracted pose is given by a skeleton composed of 17 different keypoints, each of them labeled with *x*, *y*, and *z* coordinates.

We use these skeleton keypoints (c.f. Figure 3a) to create synthetic radar data using the analytical FMCW radar model from Chen [10], which in turn is based on a human model proposed by Boulic et al. [25]. As depicted in Figure 5, This human model represents subjects with K=14 ellipsoids, each of which spans over two keypoints and represents a human limb or body part (head, torso, forearm, etc.). For each ellipsoid *k* we calculate its distance $d_{k,t}$ to the radar sensor and its corresponding Radar Cross Section (RCS) $A_{k,t}$ at any given point in time *t*, which we use to obtain the returned and mixed-down signal for a single chirp as
(3)$$s\left(t\right)=\sum_{k=1}^{K}\sqrt{\frac{A_{k,t}}{L_{k,t}}}\sin\left(2\pi f_{k,t}t+\phi_{k,t}\right),$$
with a frequency $f_{k,t}=2d_{k,t}B/T_{c}c$, an initial phase $\phi_{k,t}=4\pi f_{c}d_{k,t}/c$ and a free-space path loss $L_{k,t}={\left(4\pi d_{k,t}f_{c}/c\right)}^{2}$ for given bandwidth *B*, chirp time $T_{c}$ and lower frequency $f_{c}$. To determine $A_{k,t}$ we use the same model as in [10].

We apply this procedure throughout all FMCW chirps, thus producing a synthetic IF signal that ideally corresponds to the sampled signal from the radar sensor. We then process the synthetic signal in the same way as the real one to extract its RDMs (c.f. Figure 3b), range, and Doppler spectrograms respectively.

## 3. Method

As explained in Section 2, our pre-processed real data *x* comprises a tuple of range and Doppler spectrograms, $x^{R}$ and $x^{D}$, i.e.,:(4)$$x=\left(x^{R},x^{D}\right),\;x^{R},x^{D}\in R^{M\times N}.$$

The same holds for our synthetic data *y*, so that $y=\left(y^{R},y^{D}\right)$ with identical dimensions. In order to leverage both range and Doppler data for classification, we make use of the multi-branch CNN from [9], whose implementation details are presented in Figure 6. This CNN passes $x^{R}$ and $x^{D}$ through the convolutional branches and feeds the result of both branches into its fully connected layers. We write $\phi^{R}:\;R^{M\times N}\mapsto R^{P}$ and $\phi^{D}:\;R^{M\times N}\mapsto R^{P}$ to indicate the transformations from spectrogram to features that the range and Doppler convolutional branches respectively apply; the resulting range and Doppler feature vectors have a length of P=2304. Likewise, we use $f:\;R^{2P}\mapsto A$ to denote the transformation from range and Doppler features to the predicted activity $\tilde{a}$ through the fully connected layers. In summary, the relationship between $x^{R}$, $x^{D}$ and $\tilde{a}$ is given by
(5)$$\tilde{a}=f\left(\phi^{R}\left(x^{R}\right),\phi^{D}\left(x^{D}\right)\right),\;\tilde{a}\in A\;.$$

The architecture of our proposed Radar Activity Classification with Perceptual Image Transformation (RACPIT) is completed by appending a pair of Image Transformation Networks (ITNs) to the input of the CNN, as depicted in Figure 7. The ITNs transform real data *x* into synthetic-like data $\hat{y}=\left({\hat{y}}^{R},{\hat{y}}^{D}\right)$, where ${\hat{y}}^{R}=\psi^{R}\left(x^{R}\right)$, ${\hat{y}}^{D}=\psi^{D}\left(x^{D}\right)$ and the functions $\psi^{R}:\;R^{M\times N}\mapsto R^{M\times N}$ and $\psi^{D}:\;R^{M\times N}\mapsto R^{M\times N}$ represent the image transformations that the range and Doppler data undergo, respectively. For the ITNs we use residual convolutional autoencoders ([26], Chapter 14) with the same architecture details as [11].

Here, we first train the CNN using labeled synthetic data *y* from source and target recordings. This is required since we apply perceptual loss for the training of $\psi^{R}$ and $\psi^{D}$ [11]. Perceptual loss uses the pre-trained CNN to compare the high-level perceptual and semantic features of the transformed data $\hat{y}$ to the synthetic data *y*, which here serves as the ground truth. Once the CNN is trained, we freeze all of its layers and use the convolutional branches $\phi^{R}$ and $\phi^{D}$ to train the ITNs $\psi^{R}$ and $\psi^{D}$ with the following objective function:(6)$$\ell_{feat}^{\phi }=\frac{1}{2P}\left({∥\phi^{R}\left({\hat{y}}^{R}\right)-\phi^{R}\left(y^{R}\right)∥}_{2}^{2}+{∥\phi^{D}\left({\hat{y}}^{D}\right)-\phi^{D}\left(y^{D}\right)∥}_{2}^{2}\right)\;.$$

In that sense, instead of enforcing pixelwise resemblance, the ITNs try to generate similar feature representations extracted by the perceptual network, e.g., the CNN [11].

### Implementation and Training

RACPIT has been written on PyTorch [27] from Daniel Yang’s implementation of perceptual loss. The code is publicly available at https://github.com/fraunhoferhhi/racpit (accessed on 14 February 2022). We have trained the full pipeline (CNN and ITNs) independently for every configuration in Table 1 and we have repeated each training experiment 5 times. The ITNs have been trained for 500 epochs; other than that we have retained most of the hyperparameter values in [11], including a batch size of 4 and Adam optimization with a learning rate of $1\;\times \;10^{-4}$. As their recommendation, we have kept the total variation regularization with a strength of $1\;\times \;10^{-7}$. Prior to that, we have pre-trained the CNN for 100 epochs with a batch size of 32, using early stopping and the same optimization method and learning rate.

Besides the training of RACPIT, we have also trained just the CNN with real data as a baseline. Likewise, these baseline experiments were repeated 5 times for every different configuration in Table 1. In total, we have performed 40 training experiments, which we have run as parallel tasks on an NVIDIA Tesla V100 SXM2 GPU with CUDA 11 and 4 GB allocated per task. The training duration of RACPIT on this hardware adds up to about 10 h per experiment on average.

## 4. Results

Once trained, the ITNs have a clearly visible denoising effect (Figure 8b,e) on their real inputs (Figure 8a,d). This is to be expected, since its target is the synthetic data that we create according to the noiseless model in Equation (Equation 3) (Figure 8c,f).

We test the full pipeline on our target domain T. In order to assess the resulting accuracy in a quantitative way, we compare it to a baseline consisting of the CNN in Figure 6 trained only using real source data and tested only on the real target data. Figure 9 shows that RACPIT improves the accuracy for all configurations by 10% to 20%.

Despite this improvement, the accuracy does not reach the 90% figures of previous works [9,12,13]. A reason for this can indeed be found in the denoising behavior of the ITNs, which not only filter out background noise but also some micro-Doppler features, as it can be seen in Figure 8. In any case, the results suggest that real to synthetic transformation is an interesting strategy to explore with room for improvement. Besides the accuracy, we have also computed the average F1 score [28] and average balanced accuracy [29] per configuration in Table 2 and Table 3, respectively. These metrics indeed confirm the performance increment of RACPIT respect to the baseline.

Figure 9 also includes the results when synthetic data are fed to our baseline, which we had trained with real data. The poor accuracy, which goes as low as 20%, provides a quantitative confirmation of the dissimilarities between real and synthetic data and justifies the use of ITNs to bridge those.

## 5. Conclusions

In this paper, we have focused on improvements of deep learning methods for human activity classification on radar images. We have presented our own architecture, Radar Activity Classification with Perceptual Image Transformation (RACPIT), to mitigate the dependency of classification accuracy (and thus increase generalization) of Convolutional Neural Networks (CNNs) on the dataset recorded for the training process. RACPIT tackles this using Image Transformation Networks (ITNs) in combination with synthetic radar data, which have been generated from a radar model using human motion data. Five different activities performed by two different test subjects have been recorded in an office-like lab. The measurements have been acquired by four Frequency-Modulated Continuous-Wave (FMCW) radar sensors located at the same position, operating at around 60 GHz with different configuration settings like bandwidth, chirp time, etc. For each sensor, about 50 min of radar and video footage recordings have been taken.

We have observed that the test subjects performed the activities quite differently across recordings, considering that these were interleaved with short pauses. Hence, the data deviated largely depending on the recording time. Accordingly, the resulting radar data and the pre-processed radar images diverge also largely from each other. The recordings were thus split into a source and a target dataset with different recording times. A large decrease in classification accuracy was observed for the CNN that had been trained with the source dataset upon testing it with the target dataset.

We have tried to solve this with RACPIT in the following manner. First, we have moved on from real to synthetic data for the training of our CNN to also incorporate the target recordings into it. Since we use a lightweight radar model for data synthesis, synthetic data present a large domain shift with respect to real data, which has been confirmed in our experiments. This has inspired us to include a next step where we have investigated the impact of ITNs on the real-synthetic domain shift. As such, we have used real and synthetic data from the source domain to train two ITNs that take real data on the input and transform them so that they resemble synthetic data. By including these synthetic data from the target domain in the training, we have found an increase in classification accuracy by up to 20%. Still, the classification accuracy performance remains about 15% lower than if the CNN had been trained using both source and target datasets with real radar data.

In future research, a further in-depth investigation could show if more advanced radar models can further increase classification accuracy. For this, Shooting and Bouncing Rays (SBR) or even full 3D solver techniques could be used to generate more sophisticated radar data from human motion models. These kinematic models could be improved by using methods like dense pose algorithms [30] to generate a fine mesh of the moving human body. Finally, RACPIT should be tested and verified further by using other datasets and activities, thus human motion models that were generated purely from camera data taken in another environment, e.g., another lab with different test subjects.

## Figures and Tables

**Figure 1 sensors-22-01519-f001:**
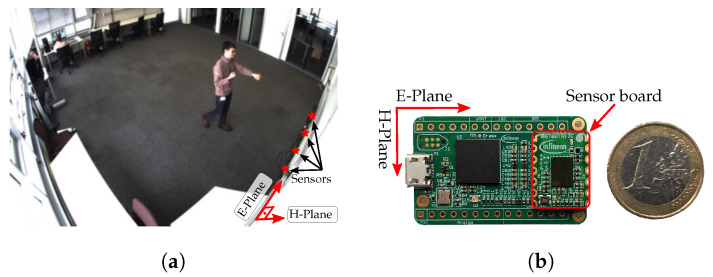
Experimental setup. (**a**) Overview of the measuring room. (**b**) Detail of the BGT60TR13C 60 GHz radar system.

**Figure 2 sensors-22-01519-f002:**
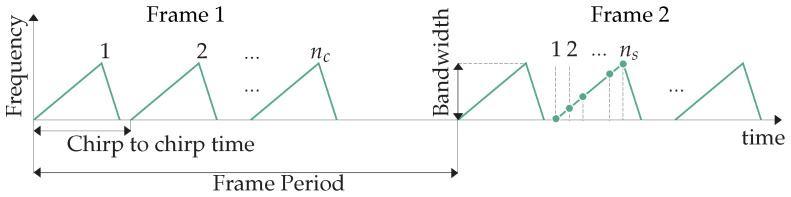
FMCW modulation and sampling pattern.

**Figure 3 sensors-22-01519-f003:**
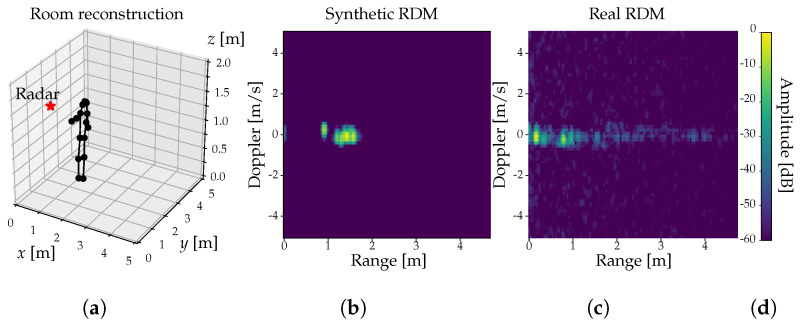
Radar data synthesis (**a**) CV-generated skeleton keypoints. (**b**) Synthetic radar data. (**c**) Real radar data. (**d**) Color bar.

**Figure 4 sensors-22-01519-f004:**
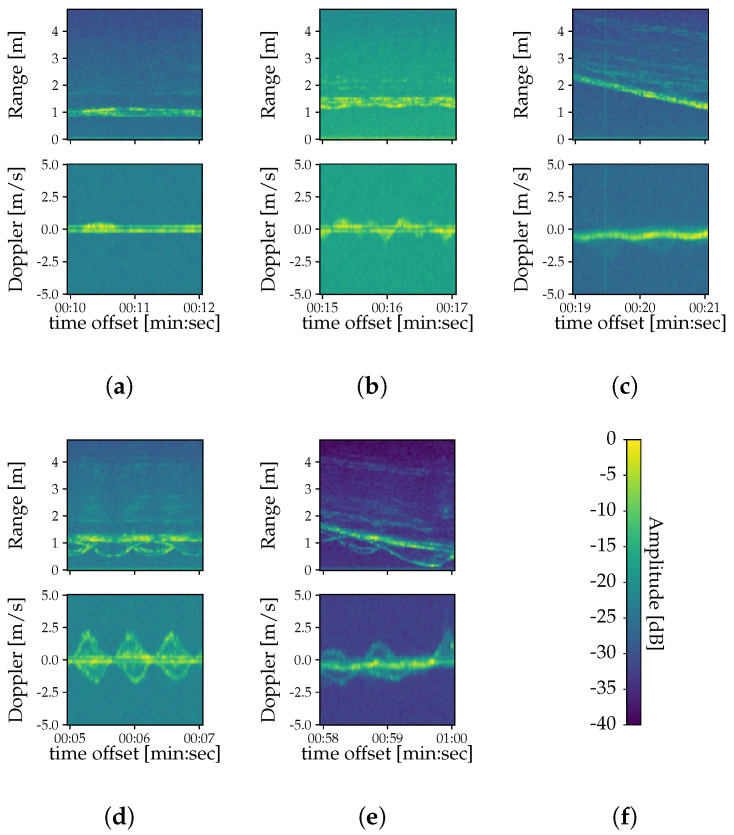
Exemplary range and Doppler spectrograms showing human activities (**a**–**e**) for configuration II. (**a**) Standing. (**b**) Waving. (**c**) Walking. (**d**) Boxing. (**e**) Boxing and walking. (**f**) Color bar showing amplitude levels in dB.

**Figure 5 sensors-22-01519-f005:**
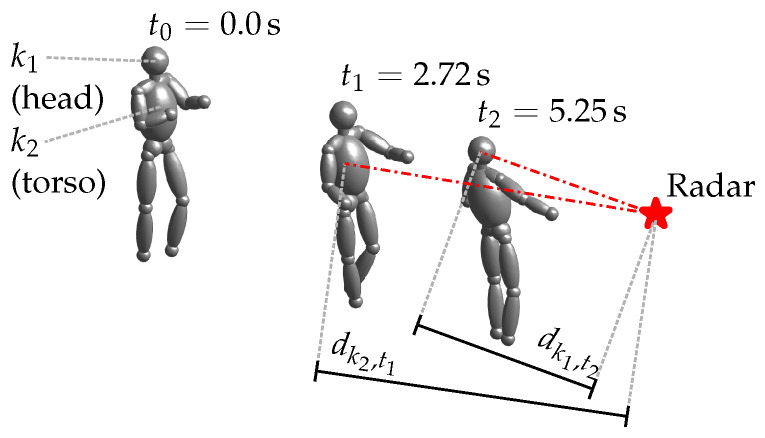
Human posture with respect to the radar at different times, as modeled by Boulic et al. [25].

**Figure 6 sensors-22-01519-f006:**
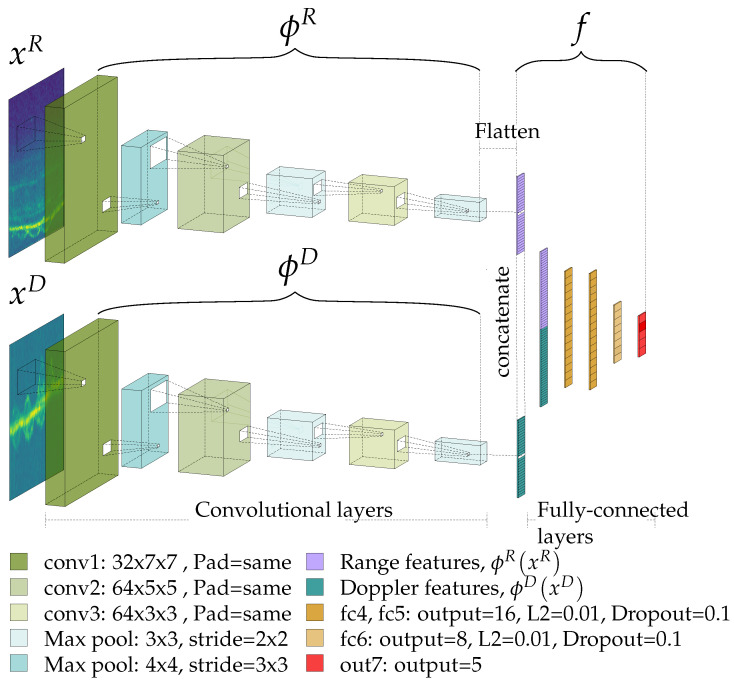
CNN architecture details.

**Figure 7 sensors-22-01519-f007:**
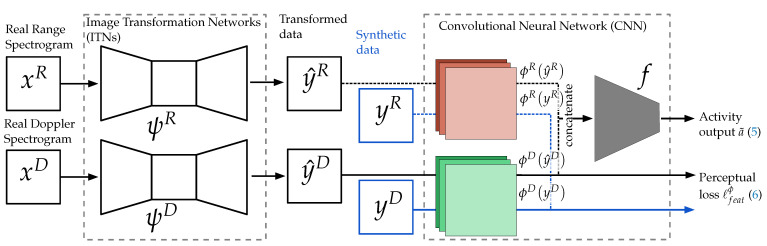
Architecture of RACPIT.

**Figure 8 sensors-22-01519-f008:**
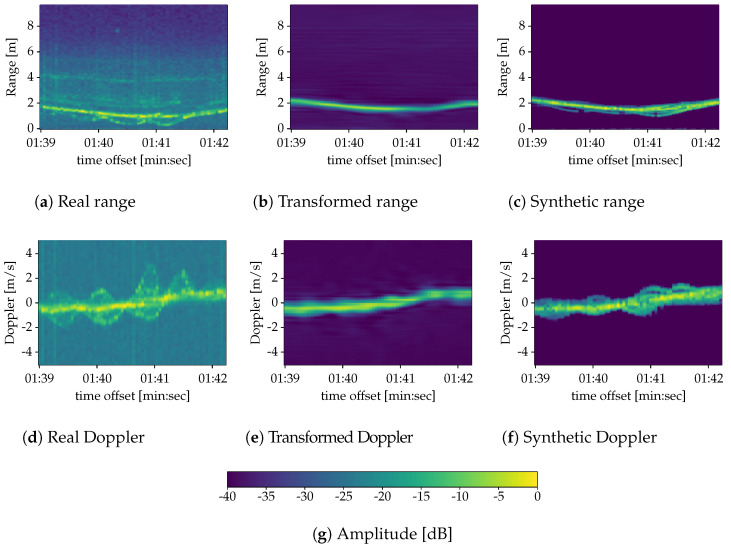
Comparison of range (**a**–**c**) and Doppler (**d**–**f**) spectrograms for real (**a**,**d**), transformed (**b**,**e**) and synthetic (**c**,**f**) data. The spectrograms display an instance of the activity item (e) in Section 2 Boxing and walking.

**Figure 9 sensors-22-01519-f009:**
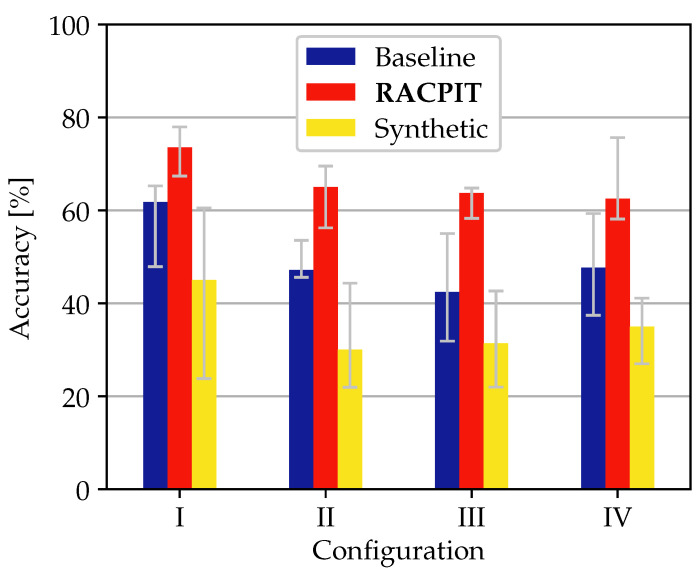
Experiment results. Each bar represents the median value out of five experiments, with the error bars indicating the minimum and maximum observed value. Besides the accuracy for the baseline and RACPIT, we also present the obtained accuracy when we directly feed synthetic data to the baseline.

**Table 1 sensors-22-01519-t001:** FMCW settings.

Configuration Name		I	II	III	IV
Chirps per frame	$n_{c}$	64	64	64	128
Samples per chirp	$n_{s}$	256	256	256	256
Chirp to chirp time	[μs]	250	250	250	250
Bandwidth	[GHz]	2	4	4	4
Frame period	[ms]	50	32	50	50
Range resolution	[cm]	7.5	3.8	3.8	3.8
Max. range	[m]	9.6	4.8	4.8	4.8
Max. speed	[m/s]	5.0	5.0	5.0	5.0
Speed resolution	[m/s]	0.15	0.15	0.15	0.08

**Table 2 sensors-22-01519-t002:** F1 score of the baseline and RACPIT for all different configurations.

Configuration	I	II	III	IV
Baseline	0.49	0.41	0.37	0.47
**RACPIT**	0.65	0.55	0.56	0.55

**Table 3 sensors-22-01519-t003:** Balanced accuracy of the baseline and RACPIT for all different configurations.

Configuration	I	II	III	IV
Baseline	0.53	0.44	0.40	0.48
**RACPIT**	0.67	0.57	0.57	0.58

## Data Availability

The data analyzed in this study are property of Infineon Technologies AG and partially available on request from the corresponding author. The data are not publicly available due to privacy issues with Infineon and due to its internal General Data Protection Regulation (in the case of video data). Code and Appendix A are publicly available at https://github.com/fraunhoferhhi/racpit, accessed on 14 February 2022.

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
