# Peer review of "Improving Radar Human Activity Classification Using Synthetic Data with Image Transformation"

_sensors, 2022, doi:10.3390/s22041519_

Round 1
Reviewer 1 Report
In this paper, the authors present an approach, which they call Radar Activity Classification with Perceptual Image Transformation (RACPIT), to do so:
First, they show that the data in their dataset differ significantly from a source time interval and a target time interval. Next, as a baseline, they train a convolutional neural network (CNN) with only real data from a source time interval and test the trained CNN with data from a target time interval.
The authors demonstrated the good performance of the new approach. To contribute to the reproducibility of their approach, authors should include the specifications of the hardware used for the experiments. Example: 16 RAM, Nvidia GeForce RTX 2080 Ti, etc.
Reviewer 2 Report
To address the challenge for the human activity sensiong caused by inconsistent measurement times, meaning the same person performing a specific 36 activity task executes the activity differently at different times, e.g. after a coffee 37 break or in a different distance to the sensor, etc.
The major issue of this paper is their lacking of the experiments. For example, the abalations. Also the efficiency of the synthetic
transformation method needs to be in-depth discovered.
The manor issues are as below,
1.line 108 i.e. should be i.e.,;
2.line 121 there is no l_i appears
and there are some typos which should be checked carefully.
Reviewer 3 Report
In this paper, the author did a significant contribution to radar human activity classification. It is quite interesting to readers. I also understand that the authors can not release the data due to some policy. Could the authors conduct experiments on an available dataset, if there is. Otherwise, can the author release a part of the dataset? Readers can follow this work smoothly.
Besides, in this paper, the author did a significant contribution to radar human activity classification. It is quite interesting to readers. I have the following comments:
1. I understand that the authors can not release the data due to some policy. Could the authors conduct experiments on an available dataset, if there is. Otherwise, can the author release a part of the dataset? Readers can follow this work smoothly.
2. The details on model training and data processing pipeline should be introduced.
3. The deployment of the trained model should be introduced.
Round 2
Reviewer 2 Report
The improvement is apparent. But,
1)The expressions for phi_d phi_r psai_d psai_r on the Figure 6,7 are confusing and not inconsistent. The figure 6 is not well organized. Redrawing is suggested.
2) The comparison is still no convincing.
Author Response
Dear Reviewer 2,
thank you for your feedback.
Regarding your comments in 1) about the math notation and Figures 6 & 7, we have undertaken the following changes:
- We have specified whether the mathematical expressions in Section 3 are vectors or functions by explicitly declaring the functions and the corresponding vector spaces (Lines 178-198)
- We have made some minor changes to Figures 6 and 7 to better indicate the specific architecture blocks that the mathematical expressions refer to.
We hope this eliminates any confusion. Should that not be the case, please tell us which specific changes would help us to do so.
As per 2), we have included the additional metrics of F1 score and balanced accuracy in Tables 2 and 3. These metrics follow the same trend as the plain accuracy (that is, an improvement of 10% to 20%) and thus they support our results.
Best regards
Reviewer 3 Report
The author solved all my concerns and this paper can be published.
Author Response
Dear Reviewer 3, thank you again for your useful feedback and your efforts.